# Effectiveness and Safety of Ultrasound-Guided Local Paricalcitol Injection in Treating Secondary Hyperparathyroidism in ESRD: A Retrospective Study

**DOI:** 10.3390/jcm11226860

**Published:** 2022-11-21

**Authors:** Shuqin Xie, Yuan Yu, Yi Liu, Siliang Zhang, Shiyi Yuan, Kui Fan, Bin Tang, Qin Zhou, Yuqing Sun, Rui Liu, Dan Cao, Yong Chen, Yelei Wang, Guangjun Liu, Huan Ma, Chenghui Tao, Li Zeng, Ling Zhong

**Affiliations:** 1Department of Nephrology, Second Affiliated Hospital, Chongqing Medical University, Chongqing 400010, China; 2Department of Nephrology, Yongchuan People’s Hospital of Chongqing, Chongqing 402160, China; 3Department of Nephrology, Dianjiang People’s Hospital of Chongqing, Chongqing 408300, China; 4The Department of Nephrology, The Fifth Hospital of Chongqing, Chongqing 400062, China; 5Department of Nephrology and Hematopathology, Second Affiliated Hospital at Fengjie, Chongqing Medical University, Chongqing 404600, China; 6Department of Nephrology, People’s Hospital of Pengshui County, Chongqing 409600, China; 7Department of Nephrology, People’s Hospital of Shizhu County, Chongqing 409100, China; 8Department of Nephrology, Fengdu People’s Hospital, Chongqing 408200, China

**Keywords:** local paricalcitol injection (LPI), intravenously paricalcitol (IP), secondary hyperparathyroidism (SHPT), end stage renal disease (ESRD)

## Abstract

Purpose: To compare the safety and efficacy of percutaneous paricalcitol injection with intravenously administered paricalcitol in treating parathyroid hyperplasia in patients with secondary hyperparathyroidism (SHPT). Methods: This study was approved by the Ethics Committee of our institution. We retrospectively collected data on patients who received percutaneous paricalcitol injection (24 patients) and intravenously administered paricalcitol (22 patients) based on their intact parathyroid hormone (iPTH) level. Serum iPTH, calcium, phosphorus, and the volume of the parathyroid gland were measured at several indicated time points after treatment, and adverse events associated with the two treatments were evaluated. Results: After 6 months of follow-up, we found that patients from the percutaneous injection group had significantly decreased levels of iPTH (from 1887.81 ± 726.81 pg/mL to 631.06 ± 393.06 pg/mL), phosphate (from 1.94 ± 0.36 mmol/L to 1.71 ± 0.34 mmol/L), and volume of the parathyroid gland (from 0.87 ± 0.50 cm^3^ to 0.60 ± 0.36 cm^3^), with relief from ostealgia within 48–72 h. In the intravenously administered group, the levels of iPTH decreased from 686.87 ± 260.44 pg/mL to 388.47 ± 167.36 pg/mL; while there was no significant change in phosphate levels, the volume of the parathyroid gland and ostealgia relief were observed at the end of follow-up. The serum calcium level did not significantly change, and no severe complications were observed in both groups. In vitro fluorescence-activated single cell sorting (FACS) analysis indicated that paricalcitol induced parathyroid cell apoptosis in a dose-dependent manner. Conclusions: Percutaneous paricalcitol injection is a selective treatment for SHPT in ESRD.

## 1. Introduction

Secondary hyperparathyroidism (SHPT) is a common complication in chronic kidney disease (CKD), affecting approximately one in three patients undergoing maintenance hemodialysis [1]. Parathyroid (PT) hyperplasia and the over-secretion of intact parathyroid hormone (iPTH) are the main characteristics of SHPT, which accelerate phosphate and calcium dysregulation, leading to debilitation by fractures and increased mortality risk in patients undergoing hemodialysis [2].

The management of SHPT remains challenging. Renal transplant is an optimal treatment; however, renal resource shortage limits its usage. Novel agents, such as calcimimetics cinacalcet and etelcalcetide, have been shown to have significant effects on PTH, calcium, and phosphate, which are commonly used in patients undergoing dialysis [3,4]. However, many patients do not adequately respond to the agent and are ultimately referred to a parathyroid surgeon to control the SHPT.

Parathyroidectomy (PTX) is a traditional surgery that has been shown to improve bone density, alleviate symptoms, and reduce cardiovascular and all-cause mortality [5]. However, intolerance to surgical anesthesia and hypoparathyroidism have limited its application in clinical practice [6]. Local medicine injection is widely used to increase therapy efficiency and decrease complications. Percutaneous ethanol injection (PEI) was introduced for the ablation of the parathyroid glands in SHPT when surgery is contraindicated or problematic [7]. However, its application in solid nodules is limited by poor seepage rate, high recurrence rate within one year of treatment, and serious adverse events, such as vocal cord paresis (VCP), thrombosis of the ipsilateral jugular vein, adhesions and necrosis of the injection area [8,9].

The vitamin D receptor (VDR) is a major regulator of PTH secretion, which is abundantly expressed in the parathyroid glands [10]. Paricalcitol is a synthetic vitamin D analog that acts as a selective activator of vitamin D receptors, and is commonly used to prevent and treat secondary SHPT in patients on maintenance hemodialysis [11]. Despite the usage of the vitamin D active and new phosphate binders in SHPT, the incidence of SHPT continues to increase with the prolonged survival time of dialysis patients [12]. Therefore, the new therapy strategy is often necessary for patients suffering from severe drug-resistant SHPT.

In this study, we investigated whether percutaneous local injection (LPI) in the hyperplastic parathyroid gland could be used to treat hyperparathyroidism in chronic kidney disease by exploring its efficacy, safety, and complications.

## 2. Methods

### 2.1. Ethics Statement

This study was approved by the Ethics Committee of the Second Affiliated Hospital of Chongqing Medical University (No.2022(1)). Ethical approval exempted the need for informed consent.

### 2.2. Subjects

The data of 46 end-stage renal disease (ESRD) patients (26 males and 20 females) who underwent regular hemodialysis for more than one year and dialyzed 3 times per week in 7 blood purification centers from February 2020 to September 2021 were assessed. The study protocol was reviewed and approved by the Ethics Committee of the Second Affiliated Hospital of Chongqing Medical University and was performed in accordance with the Declaration of Helsinki. This retrospective and observational study was exempted from the need for written informed consent.

The study inclusion criteria were: (1) age, 18–80 years; (2) iPTH ≥ 500 ng/L; (3) uncontrolled SHPT with adequate oral medication therapy; (4) hyperplastic, at least one parathyroid gland’s diameter over 0.6 cm detected by ultrasound; and (5) no severe bleeding disorders, cardiac insufficiency, or uncontrollable hypertension. The exclusion criteria were: (1) abnormal coagulation function tests (prothrombin time > 25 s, prothrombin activity < 40% and platelet count < 100 × 10^9^/L); (2) severe cardiopulmonary dysfunction; (3) hypercalcemia; (4) vitamin D intoxication; and (5) ongoing pregnancy.

### 2.3. Ultrasound-Guided Local Paricalcitol Injection (US-Guided LPI)

US-guided LPI ablation was performed with patients who were refractory to medical therapy for at least 6 months, including intravenous paricalcitol treatment. The treatment was administered in the supine position and mild neck extension. Each nodule volume was calculated using the equation: V= π abc/6, as previously reported; where V represents volume, a represents the largest diameter, and b and c represent the other two diameters [8]. Then, the puncture site was anesthetized with 2% lidocaine, followed by the insertion of an 18-gauge needle into the center of the nodule and slow injection of 2 mL/cm^3^ paricalcitol based on the nodule volume, and immerse the gland totally. Following the procedure, every gland’s diameter over 0.6 cm was treated, and each patient was observed for 24 h in the hospital. No other therapies for SHPT were administered concurrently.

### 2.4. Intravenously Administered Paricalcitol 

The dose of paricalcitol was based on the Summary of Product Characteristics of Hospira S.P.A. Briefly, paricalcitol was administered intravenously to the bloodline at 0.1 ug/kg every dialysis session, and the iPTH serum concentration level was monitored monthly. The paricalcitol dose was reduced when the iPTH level was lower than 150 pg/mL [13,14]. All the patients did not take extra medical intervention for SHPT during the follow-up.

### 2.5. Laboratory Tests and Follow-Up

All the patients’ demography, primary disease, and dialysis history were recorded. The patient-reported symptoms of ostealgia, arthralgia, and cutaneous pruritus were assessed with a questionnaire before and after treatment. Blood tests for PTH, calcium, and phosphorus were conducted 24 h before and one week after every therapy session, and at the end of follow-up in the PIP group. In the intravenous paricalcitol (IP) group, blood test performance was determined based on the 24 h before the first time of treatment and at the end of follow-up. The total follow-up time was 6 months. All the data were obtained from the same laboratory at the Second Affiliated Hospital of Chongqing Medical University. 

### 2.6. Human Parathyroid Gland Isolation and Ex Vivo Culture

Human PT tissues were obtained from 3 ESRD patients who failed to respond to pharmacological therapy, including calcimimetics, calcitriol, or vitamin D analog, or a combination of calcimimetics with calcitriol or vitamin D analogues. All the patients underwent PTX due to SHPT at the Second Affiliated Hospital of Chongqing Medical University after providing signed informed consent. The glands from different patients were handled separately and PT cells from different patients were not mixed. The parathyroid cells were isolated and cultured as previously described with minor modification [2]. Briefly, the parathyroid specimen was transported in ice-cold Hanks’ balanced salt solution (Invitrogen) containing CaCl_2_ and micro-dissected in PBS. The glands were digested at 37 °C for 20 min in a 1640 culture medium (Zhong Qiao Xin Zhou Biotechnology Co., Shanghai, China) containing 1 mg/mL collagenase (Type I, Worthington, Franklin County, OH, USA) and 0.2 mg/mL DNase I (Type IV, Sigma, St. Louis, MI, USA). The cloudy suspension was centrifuged at 1000× *g* rpm for 5 min and the deposit was collected. The pellet was resuspended with 5 mL 1640 culture medium and filtered with a 100-μm pore-size nylon filter. PT cells were repeatedly washed thrice in a 1640 culture medium and suspended in a 1640 culture medium supplemented with 10% bovine serum albumin, 1% penicillin G, and 1% streptomycin. The cells were incubated in a humidified atmosphere of 95% air/5% CO_2_ at 37 °C. The medium was renewed every other day. To examine the effects of paricalcitol in PT cells, we treated the PT cells with paricalcitol (0, 0.1, 0.2, 0.4, 0.8, 1.6, and 3.2 mg/mL) in a culture medium for 48 h. To exclude the effects of paricalcitol excipients, we used the same volume of 20% ethanol as the control. 

### 2.7. Apoptosis Analysis

Apoptosis was measured by flow cytometry using an Annexin V-FITC apoptosis detection kit (BD Bioscience, Franklin Lakes, NJ, USA). Annexin V-FITC staining was used to detect early-stage apoptosis. Necrotic or late-stage apoptotic cells were labeled with propidium iodide. The number of cells labeled with Annexin V-FITC and propidium iodide was quantified using the fluorescence-activated single cell sorting (FACS) caliber flow cytometer, and the data were analyzed with CellQuest software (BD Biosciences, Version 5.1).

### 2.8. Statistical Analysis

Statistical analyses were performed using GraphPad Prism 9. Continuous data are presented as means ± standard deviation. Continuous variables were analyzed using the Student *t*-test or the Wilcoxon rank-sum test, as applicable. Paired-sample *t*-tests and paired sample Wilcoxon signed-rank tests were used to compare pre- and post-ablation treatment outcomes. The relationships between two variables were calculated using Spearman rank correlation analysis. *p* values < 0.05 were used to indicate a significant difference. 

## 3. Results

### 3.1. Participant Characteristics

The study enrolled 46 patients on maintenance hemodialysis from 7 blood purification centers. Table 1 shows the baseline clinical characteristics of the enrolled patients. The data showed that all variables were well-balanced between the two groups (*p* > 0.05). The dosage used of paricalcitol in two groups is shown in Appendix A.

### 3.2. Outcomes

Following the 6-month follow-up, no patients had died. In the IP group, the level of iPTH decreased from 686.87 ± 260.44 pg/mL to 388.47 ± 167.36 pg/mL at the end of follow-up. In the LPI group, iPTH decreased from 1887.81 ± 726.81 pg/mL to 1061.77 ± 701.49 pg/mL within 1 week after the first session of LPI. The iPTH level of 17 patients rebounded above 600 pg/mL in a month and went through the second LPI treatment. The iPTH decreased from 1625.81 ± 795.69 pg/mL to 876.12 ± 501.01 pg/mL after the second session of LPI treatment in 1 week. The iPTH level of eight patients rebounded in 3 months, and they were given a third session of LPI treatment, following which their iPTH level was then observed to decrease from 1208.89 ± 322.06 pg/mL to 831.32 ± 280.74 pg/mL in 1 week. At the end of follow-up, the iPTH level of all the patients decreased from 1887.81 ± 726.81 pg/mL to 631.06 ± 393.06 pg/mL, and the patients underwent 2.04 sessions of treatment. No significant change in serum calcium level after 6 months of follow-up in both groups was observed. The serum phosphate level of the IP group did not significantly change at the end of follow-up, while a significant decrease in phosphate level in the LPI group from 1.94 ± 0.36 mmol/L to 1.71 ± 0.34 mmol/L was observed (Table 2).

Ostealgia, arthralgia, and cutaneous pruritus are the most common symptoms of SHPT. During follow-up, we observed a clear improvement in SHPT symptoms after LPI treatment (Table 3). Ostealgia occurred in 21 of 24 patients (87.5%) before LPI treatment. After the first session, the reported symptoms had remitted in 19 cases (90.5%) within 24 h to 48 h. Three patients (12.5%) reported arthralgia before LPI, of whom remission was reported in two patients (66.6%) in 2 months. Two patients (8.33%) had symptoms of pruritus before LPI, which was relieved in one (50%) at the end of the follow-up. In total, 9 of 22 patients (40.9%) reported ostealgia, 2 (9.09%) had arthralgia, and 2 (9.09%) had pruritus separately in the IP group before treatment. In addition, at the end of follow-up, one patient (11.11%) had remission from ostealgia; however, none reported relief from arthralgia and pruritus.

### 3.3. Volume Change of Parathyroid Gland after Therapy

Ultrasound and CT examination indicated a significant decrease in gland volume after LPI treatment (Figure 1A–D). In the LPI group, the parathyroid gland volume had decreased from 0.87 ± 0.50 cm^3^ to 0.60 ± 0.36 cm^3^ (*p* = 0.02); however, it did not significantly change in the IP group (from 0.79 ± 0.30 cm^3^ to 0.88 ± 0.28 cm^3^, *p* = 0.83) during follow-up. Comparatively, the gland volume in the PIP group was significantly decreased compared to the IP group (*p* = 0.04) (Figure 1E).

### 3.4. Complications

As shown in Table 4, the number of adverse events in the LPI group (5, 20.83%) was higher than in the IP group (4, 14.28%). After LPI treatment, five patients reported facial numbness (1, 4.17%), hoarseness (3, 12.5%), and cough (1, 4.17%), which remitted within 3 days without medical intervention. Four patients had conjunctivitis (2, 7.14%), palpitation (1, 3.57%), and subcutaneous induration (1, 3.57%) in the IP group, which spontaneously disappeared in 3 days.

### 3.5. Paricalcitol Induced Apoptosis

To explore the functional effects of paricalcitol in shrinking the parathyroid gland, FACS was used to evaluate the effects of different concentrations on the apoptosis rates of parathyroid cells (PT cells), and the same volume of 20% ethanol, the excipient of the paricalcitol, as the control group. The results showed that the excipient induced apoptosis following an increase in ethanol concentration. However, the apoptosis rate was significantly increased to 75.59%, 83.32%, and 84.81% by 0.8 μg/mL, 1.6 μg/mL, and 3.2 μg/mL paricalcitol in 48 h, respectively, while that of the control group was 29.54%, 34.5%, and 54.99% (Figure 2), indicating that paricalcitol induced apoptosis in PT cells in a dose-dependent manner.

## 4. Discussion

Paricalcitol functions as a selective activator of VDR and has been shown to inhibit iPTH secretion from the parathyroid gland in SHPT. It can be intravenously administered on alternate days and used from 0.4 to 0.24 μg/kg [15]. However, paricalcitol can also increase calcium and phosphorus absorption from the intestines, increasing the risk of hypercalcemia and the progression of cardiovascular calcification in ESRD [16]. Loco-regional treatment was shown to have superior therapeutic effects and fewer adverse events in clinics. Previous research has shown that direct injection of calcitriol, a non-selective vitamin D receptor activator, decreased serum iPTH concentration, gland size, and did not show serious adverse effects in a 3 month follow-up in nine patients.

This study found that local paricalcitol injection could effectively treat secondary hyperparathyroidism in ESRD. The LPI treatment decreased iPTH significantly without severe adverse events in SHPT patients with ESRD. The recurrence rate of LPI was about 70.8%, which was not inferior to subtotal PTX (5–80%) [17,18] and ethanol injection (80%) [19] in SHPT. After an average of 2.04 treatment sessions, a significant reduction in the volume of the parathyroid gland was observed. Moreover, similar to microwave and radiofrequency ablation [20,21], LPI showed a similar remission rate of ostealgia in the patients, indicating good clinical effects. In vitro studies showed that once the concentration was >0.8 μg/mL, the paricalcitol induced significant apoptosis of parathyroid cells, suggesting that loco-regional injection might induce parathyroid cell apoptosis by increasing the paricalcitol concentration in the parathyroid gland.

Serum calcium metabolism disorder is the most common complication after radiofrequency ablation, parathyroid resection, and peripheral usage of paricalcitol [21,22,23,24]. Hypocalcemia is a life-threatening complication that includes muscle cramps, arrhythmias, laryngospasm, and seizures [25], which requires extra calcium supplements for several months. The incidence rate of hypocalcemia ranged from 10.2% in the thermal ablation group to 24.3% in the PTX group. In our study, percutaneous paricalcitol injection did not significantly disturb the serum calcium level in short- and long-term observation.

Hyperphosphatemia has been associated with high risks of complications, including cardiovascular mortality and cardiovascular anomalies requiring hospitalization [25]. Intravenously administered paricalcitol was reported to increase the intestinal absorption of phosphorus. Unlike intravenously administered, we found that LPI was associated with a significant decrease in serum phosphorus level at the end of follow-up, suggesting that percutaneous injection of paricalcitol might reduce vascular calcification, reduce the incidence of cardiovascular events, and have a better prognosis in ESRD. At the end of the follow-up, none of the patients had cardiovascular or cerebral events. However, the long-term outcome of the LPI treatment in ESRD needs further verification.

There were several limitations in this study. First, this is a retrospective study with a relatively small sample size. Further investigations are needed to assess the long-term efficacy and outcomes of the treatment. Second, this study did not account for the difference in bone metabolism associated with LPI and IP. Third, the symptom characteristics of the two groups were different, which might have induced a certain level of bias.

In conclusion, our study provides initial evidence that a US-guided LPI of parathyroid hyperplasia might be a selective treatment, with less side effects, and has the capability for effective treatment of SHPT in ESRD; in addition, it may represent an alternative to minimally invasive surgery for patients with refractory drug-resistant SHPT or SHPT patients intolerant to parathyroidectomy, and without severe adverse events.

## Figures and Tables

**Figure 1 jcm-11-06860-f001:**
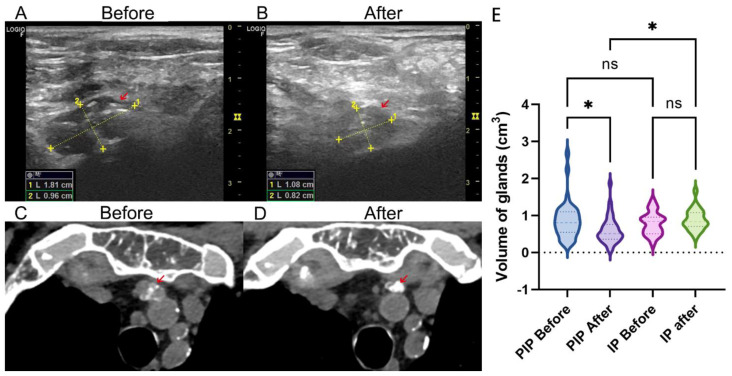
Gland volume change of the LPI group. The ultrasound image of the parathyroid gland before (**A**) and after (**B**) LPI treatment. The CT image of the parathyroid gland (red arrow) before (**C**) and after (**D**) LPI treatment. (**E**) Violet plot showing the volume change between LPI and IP treatment. * *p* < 0.05.

**Figure 2 jcm-11-06860-f002:**
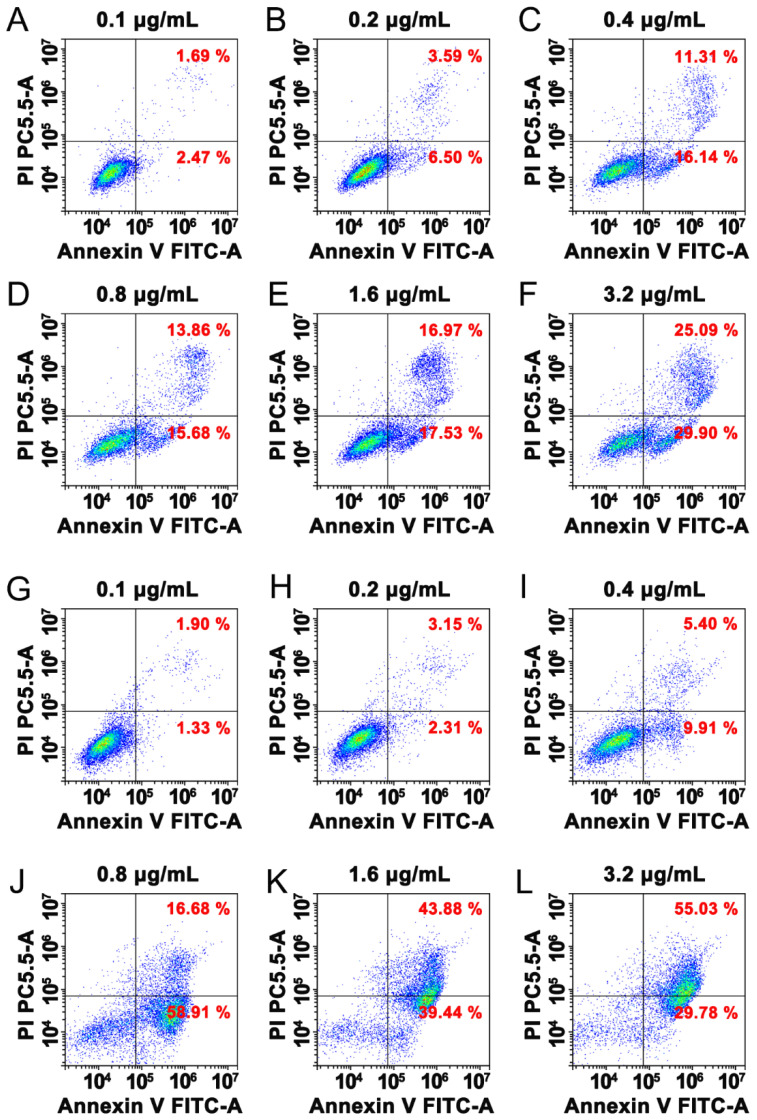
FACS analysis of paricalcitol inducing parathyroid cells apoptosis. (**A**–**F**) was treated with ethanol as control, (**G**–**L**) was treated with paricalcitol.

**Table 1 jcm-11-06860-t001:** Clinical and demographic characteristics.

	LPI Group (*n* = 24)	IP Group (*n* = 22)	*p*
Age, y	46.07 ± 12.69	51.27 ± 12.97	0.18
Sex, male, no. (%)	14, 58.33%	12, 54.55%	0.99
Primary disease type, no. (%)			0.84
Chronic nephritis	10, 41.66%	11, 50%	
Diabetes	6, 25%	5, 22.73%	
Hypetension	8, 33.33%	6, 27.27%	
Dialysis duration, mo	86.25 ± 29.95	79.85 ± 23.21	0.43
Hemoglobin	112.93 ± 17.62	116.73 ± 11.15	0.39
Preoperative medicine usage			0.40
Calcitriol	8, 33.3%	6, 27.27%	
Cinacalcet	17, 70.83%	5, 22.73%	
Phosphate binder	19, 79.17%	18, 81.82%	

**Table 2 jcm-11-06860-t002:** The efficacy of treatment.

	LPI Group							IP Group (*n* = 22)	
	Session 1 (*n* = 24)		Session 2 (*n* = 17)		Session 3 (*n* = 8)		End Follow Up (*n* = 24)	Before	After
	Before	After	Before	After	Before	After			
iPTH (pg/mL)	1887.81 ± 726.81	1061.77 ± 701.49 *	1625.81 ± 795.69	876.12 ±501.01 *	1208.89 ± 322.06	831.32 ± 280.74 *	631.06 ± 393.06 ^#^	686.87 ± 260.44	388.47 ± 167.36 ^#^
Calcium (mg/dL)	2.26 ± 0.22	2.16 ± 0.28	2.23 ± 0.29	2.15 ± 0.29	2.31 ± 0.08	2.22 ± 0.11	2.12 ± 0.23	2.36 ± 0.17	2.43 ± 0.11
Phosphate (mg/dL)	1.94 ± 0.36	1.86 ± 0.49	1.76 ± 0.41	1.84 ± 0.47	1.40 ± 0.42	1.48 ± 0.24	1.71 ± 0.34 ^#^	2.07 ± 0.40	2.06 ± 0.46

* *p* < 0.05 vs. before treatment. ^#^
*p* < 0.05 vs. first session treatment.

**Table 3 jcm-11-06860-t003:** Patients’ reported symptoms.

Clinial Symptoms	IP Group (*n* = 22)		LPI Group (*n* = 24)	
	Incident	Remission	Incident	Remission
Ostealgia	9, 40.9%	1, 11.11%	21, 87.5% **	19, 90.5% *
Arthralgia	2, 9.09%	0, 0%	3, 12.5%	2, 66.6%
Cutaneous pruritus	2, 9.09%	0, 0%	2, 8.33%	1, 50%

** means *p* < 0.01, * means *p* < 0.05.

**Table 4 jcm-11-06860-t004:** The side effects of the treatment.

Symptom	LPI Group (*n* = 24)		IP Group (*n* = 22)	
	Number	Durition Time (Day)	Number	Durition Time (Day)
Facial numbness (*n*, %)	1, 4.17%	1		
Hoarseness (*n*, %)	3, 12.5%	2.33 ± 0.57		
Cough (*n*, %)	1, 4.17%	1		
Conjunctivitis (*n*, %)			2, 9.09%	1
Palpitation (*n*, %)			1, 4.54%	0.5
Subcutaneous induration (*n*, %)			1, 4.54%	3

## Data Availability

Data is contained within the article or Appendix A.

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
