# Peer review of "Effectiveness and Safety of Ultrasound-Guided Local Paricalcitol Injection in Treating Secondary Hyperparathyroidism in ESRD: A Retrospective Study"

_jcm, 2022, doi:10.3390/jcm11226860_

Round 1

Reviewer 1 Report

1. The authors concluded in this retrospective observational study, direct paricalcitol injection in parathyroid glands might be effective treatment for managing SHPT in ESRD and may represent an alternative to minimally invasive surgery for patients with resistant to medical therapy, or intolerant to parathyroidectomy (PTx). However, the authors showed only 6-month decrease in PTH, serum P levels and improve in the SHPT-related symptoms for markedly advanced SHPT in the manuscript. The efficacy of the direct injection method of VDRA in enlarged parathyroid gland has been already reported (Nephrol Dial Transplant. 2003 Jun;18 Suppl 3: iii42-6). It has been proved that cinacalcet decrease the rate of PTx by EVOLVE study.  In the current paper, cinacalcet shows similar effect to PTx if not advanced SHPT (J Clin Endocrinol Metab. 2022 Jun 16; 107: 2016). Calcimimetics has been established as medical therapy of SHPT. The authors should make clearly the subjects and the purpose of the direct paricalcitol injection therapy.

2. The authors concluded might be effective treatment for managing SHPT in ESRD, emphasized too much because of above reasons. The authors should change conclusions according to these results.

3. This manuscript has major problems with the method,

1) The criteria for selection of the paricalcitol injection gland, largest gland or more than two glands, are not shown.  It is unknown which glands were treated in the method.

2) After injection medical therapy to prevent the recurrence are not shown.

4. Page 6, line 238. The numbers of control group were 30.34%, 36.1% and 57.38%, different in the figure 2.

Author Response

Reviewer 1

  1. The authors concluded in this retrospective observational study, direct paricalcitol injection in parathyroid glands might be effective treatment for managing SHPT in ESRD and may represent an alternative to minimally invasive surgery for patients with resistant to medical therapy, or intolerant to parathyroidectomy (PTx). However, the authors showed only 6-month decrease in PTH, serum P levels and improve in the SHPT-related symptoms for markedly advanced SHPT in the manuscript. The efficacy of the direct injection method of VDRA in enlarged parathyroid gland has been already reported (Nephrol Dial Transplant. 2003 Jun;18 Suppl 3: iii42-6). It has been proved that cinacalcet decrease the rate of PTx by EVOLVE study.  In the current paper, cinacalcet shows similar effect to PTx if not advanced SHPT (J Clin Endocrinol Metab. 2022 Jun 16; 107: 2016). Calcimimetics has been established as medical therapy of SHPT. The authors should make clearly the subjects and the purpose of the direct paricalcitol injection therapy.

 Response: Thank you for your valuable opinion. Because the patients enrolled in the local paricalcitol injection treatment displayed internal medicine resistance for at least 6 months and were intolerant or unwanted to take parathyroidectomy treatment. So, we think this alternative method may be meaningful. We add this information in line 111.

Because paricalcitol has shown a significant survival advantage compared to unselective VDRA (N Engl J Med 2003; 349:446-456), for long-term safety reasons, we evaluated the safety and efficacy of selective VDRA local injection.

  1. The authors concluded might be effective treatment for managing SHPT in ESRD, emphasized too much because of above reasons. The authors should change conclusions according to these results.

 Response: We change the conclusion in line 42 and 307.

  1. This manuscript has major problems with the method,

1) The criteria for selection of the paricalcitol injection gland, largest gland or more than two glands, are not shown.  It is unknown which glands were treated in the method.

Response: We enroll the patients with at least one gland’s diameter over 0.6 cm. If more than one gland was in a patient, we treated every gland which a diameter over 0.6 cm. We add this information in line 103 and 120. And large glands, such as a diameter over 1.2 cm were too less, we didn’t analyze them separately.

2) After injection medical therapy to prevent the recurrence are not shown.

 Response: Because we want to make sure whether the patients can consistently relieve without extra medical intervention, these patients didn’t take medical therapy to prevent the recurrence. We add this information in line 121 We can assess the combination therapy in further study.

  1. Page 6, line 238. The numbers of control group were 30.34%, 36.1% and 57.38%, different in the figure 2.

Response: Sorry for the mistake, we correct it in line 250.

Reviewer 2 Report

This manuscript offers an interesting contribution to the therapeutic options for secondary hyperparathyroidism.  However, there are several concerns which must be addressed prior to publication.

1. What is the value of the in vitro data in this study?  It is an inappropriate conclusion made in lines 262-265 that the in vitro dose response supports a difference between regional and systemic therapies.  This finding may only provide some relevance if we understood the dosing received by the PPI subjects compared to the IP subjects.  If the in vitro information is kept, then it must be specified whether these 3 subjects were different than the other 46, and it must be specified what treatment they received prior to parathyroidectomy...at the very least.

- Must add in line 40 that this is in vitro

2. Acronyms are confused and misused throughout the paper.  FACS must be defined.  PIP seems to be used interchangeably with PPI.  IP is never defined when first used in line 129.

3. No actual dose is every provided for PPI, only a volume.  Without this, the paper is unpublishable.  The average paricalcitol dose must be provided for the PPI group overall, and for the Session 1/2/3 subgroups.  The average paricalcitol dose must be provided for the IP group.  And, if the in vitro data is kept, then we must be told what paricalcitol exposure that tissue had (at least in the 6 months if not longer) prior to excision.

4. The time-frame for PTH lowering after Session 2 and 3 must be specified, as it was for session 1.

5. It must be specified if the PPI group received concurrent activated Vitamin D or calcimimetic therapy.  It must be specified if the IP group received concurrent treatments as well.  This is quite common in retrospective study and must be clarified.

6. P values must be provided for section 3.3.

7. Line 289 - Change clinical to symptom

8. It should be specified more in the methods who performed the PPI, nephrologist/endocrinologist/radiologist?

English Language Edits:

Line 34 - change relieve to relief

Line 51 - change disorder to dysregulation

Line 63-64 - delete recurrence and permanent

Line 80 - delete the

Line 83 - change ablation to treat

Line 84 - change efficiency to efficacy

Line 224 & Table 4 - what is bucking?

Line 226 - change eyeball congestion (not sure what is meant here), change flustered to palpitation

Line 227 - change nodule to induration

Table 4 - Symptom is misspelled and the terminology should match between the text and Table

Author Response

Reviewer 2

This manuscript offers an interesting contribution to the therapeutic options for secondary hyperparathyroidism.  However, there are several concerns which must be addressed prior to publication.

  1. What is the value of the in vitro data in this study?  It is an inappropriate conclusion made in lines 262-265 that the in vitro dose response supports a difference between regional and systemic therapies.  This finding may only provide some relevance if we understood the dosing received by the PPI subjects compared to the IP subjects.  If the in vitro information is kept, then it must be specified whether these 3 subjects were different than the other 46, and it must be specified what treatment they received prior to parathyroidectomy...at the very least.

Response: We delete the inappropriate conclusion in lines 262-265. But we think the vitro study may give us some information about why local injection paricalcitol showed more efficiency compared to systemic treatment, so we keep the vitro study. And we agree with you that the medical information of the 3 patients prior to PTX is important. We add this information in line 142.

- Must add in line 40 that this is in vitro

Response: We added this information.

  1. Acronyms are confused and misused throughout the paper.  FACS must be defined.  PIP seems to be used interchangeably with PPI.  IP is never defined when first used in line 129.

Response: we correct these mistakes.

  1. No actual dose is every provided for PPI, only a volume.  Without this, the paper is unpublishable.  The average paricalcitol dose must be provided for the PPI group overall, and for the Session 1/2/3 subgroups.  The average paricalcitol dose must be provided for the IP group.  And, if the in vitro data is kept, then we must be told what paricalcitol exposure that tissue had (at least in the 6 months if not longer) prior to excision.

Response: The dosage is mentioned in part 2.33 which is 1ml/cm3 in the original manuscript, however, as per your notice, we correct it to 1ml/cm3/grand in the revised manuscript in line 119, and it may be clearer. Because most patients had more than one grand, our vitro data showed that its effect depend on the local concentration of paricalcitol. To avoid insufficient treatment or overdosage treatment, we use 1ml/cm3/grand instead of volume/person to calculate the dosage.

  1. The time-frame for PTH lowering after Session 2 and 3 must be specified, as it was for session 1.

Response: We added this information in line 196 and 199.

  1. It must be specified if the PPI group received concurrent activated Vitamin D or calcimimetic therapy.  It must be specified if the IP group received concurrent treatments as well.  This is quite common in retrospective study and must be clarified.

Response: Both groups didn’t take extra medical intervention for SHPT during the follow-up, we added this information in lines 121 and 128.

  1. P values must be provided for section 3.3.

Response: We added the p values in line 224, 225 and 226.

  1. Line 289 - Change clinical to symptom

Response: We corrected it in line 304.

  1. It should be specified more in the methods who performed the PPI, nephrologist/endocrinologist/radiologist?

English Language Edits:

Line 34 - change relieve to relief

Line 51 - change disorder to dysregulation

Line 63-64 - delete recurrence and permanent

Line 80 - delete the

Line 83 - change ablation to treat

Line 84 - change efficiency to efficacy

Line 224 & Table 4 - what is bucking?

Line 226 - change eyeball congestion (not sure what is meant here), change flustered to palpitation

Line 227 - change nodule to induration

Table 4 - Symptom is misspelled and the terminology should match between the text and Table

Response: Thank you for your valuable advice. We correct the above mistake.

Round 2

Reviewer 1 Report

All the problems that I pointed out are revised appropriately.

Author Response

Thank you for your valuable opinions.

Reviewer 2 Report

The edits made are appreciated.  However, two major concerns persist.

First, the authors have still not provided adequate information regarding the dose of paricalcitol via local injection.  In Line 119, "grand" has been added, which does not make any sense and must be a spelling mistake.  Furthermore, the average doses should be compared between the study groups in table format as well.  Without this data, this study is not appropriate for publication.

Second, the inclusion of the in vitro data has not been logically connected to this research question.

Minor corrections include problems with English language and notation:

Line 86 - "efficacy" is mis-spelled

Line 112 - "had internal medicine resistance" should be changed to "were refractory to medical therapy"

Line 113 - change "taken" to "administered"

Line 121-122 - Change the sentence to "No other therapies for SHPT were administered concurrently."

Line 142 - change "fail" to "failed"

Line 145 - change SPTH to SHPT

The acronyms PIP and LPI continue to be used interchangeably, which is inappropriate for publication.  This persists in Tables 2 and 3.

Table 4 - Flustered has been kept and Palpitations crossed out - it should be the reverse.

No contractions should be used in the text narrative, example line 265.

Author Response

The edits made are appreciated.  However, two major concerns persist.

First, the authors have still not provided adequate information regarding the dose of paricalcitol via local injection.  In Line 119, "grand" has been added, which does not make any sense and must be a spelling mistake.  Furthermore, the average doses should be compared between the study groups in table format as well.  Without this data, this study is not appropriate for publication.

Response: we add the dosage we used in the supplement table and correct the above error.

Second, the inclusion of the in vitro data has not been logically connected to this research question.

Response: We hold the opinion that the vitro data is important, so we want to keep this part.

Minor corrections include problems with English language and notation:

Line 86 - "efficacy" is mis-spelled

Line 112 - "had internal medicine resistance" should be changed to "were refractory to medical therapy"

Line 113 - change "taken" to "administered"

Line 121-122 - Change the sentence to "No other therapies for SHPT were administered concurrently."

Line 142 - change "fail" to "failed"

Line 145 - change SPTH to SHPT

The acronyms PIP and LPI continue to be used interchangeably, which is inappropriate for publication.  This persists in Tables 2 and 3.

Table 4 - Flustered has been kept and Palpitations crossed out - it should be the reverse.

No contractions should be used in the text narrative, example line 265.

Response: Thank you for your kind suggestions, we correct the above mistakes.